# PSMA Expression Correlates with Improved Overall Survival and VEGF Expression in Glioblastoma

**DOI:** 10.3390/biomedicines11041148

**Published:** 2023-04-11

**Authors:** Alexander Yuile, Adrian Lee, Elizabeth A. Moon, Amanda Hudson, Marina Kastelan, Samuel Miller, David Chan, Joe Wei, Michael F. Back, Helen R. Wheeler

**Affiliations:** 1Department of Medical Oncology, Royal North Shore Hospital, Reserve Road, St Leonards, NSW 2065, Australia; 2Faculty of Medicine and Health Sciences, Northern Clinical School, The University of Sydney, Sydney, NSW 2000, Australia; 3The Brain Cancer Group, North Shore Private Hospital, Westbourne Street, St Leonards, NSW 2065, Australia; 4Department of Radiation Oncology, Royal North Shore Hospital, Reserve Road, St Leonards, NSW 2065, Australia

**Keywords:** glioblastoma, PSMA, VEGF, immunohistochemistry, angiogenesis, biomarker

## Abstract

Background: Glioblastomas are the most common and fatal primary brain malignancy in adults. There is a growing interest in identifying the molecular mechanisms of these tumors to develop novel treatments. Glioblastoma neo-angiogenesis is driven by VEGF, and another potential molecule linked to angiogenesis is PSMA. Our study suggests the potential for an association between PSMA and VEGF expression in glioblastoma neo-vasculature. Methods: Archived *IDH1/2* wild-type glioblastomas were accessed; demographic and clinical outcomes were recorded. PSMA and VEGF expression by IHC were examined. Patients were dichotomized into PSMA expression high (3+) and low (0–2+) groups. The association between PSMA and VEGF expression was evaluated using Chi^2^ analysis. OS in PSMA high and low expression groups were compared using multi-linear regression. Results: In total, 247 patients with *IDH1/2* wild-type glioblastoma with archival tumor samples (between 2009–2014) were examined. PSMA expression correlated positively with VEGF expression (*p* = 0.01). We detected a significant difference in median OS between PSMA vascular endothelial expression high and low groups—16.1 and 10.8 months, respectively (*p* = 0.02). Conclusion: We found a potential positive correlation between PSMA and VEGF expression. Secondly, we showed a potential positive correlation between PSMA expression and overall survival.

## 1. Introduction

Gliomas are the most common primary brain malignancy worldwide [1]. The most common and aggressive glioma is *IDH1/2* wild-type glioblastoma, with a median overall survival of 14 months [2]. The 2016 and 2021 WHO classifications have integrated and significantly enhanced the importance of molecular alterations in the diagnoses of gliomas. Grouping gliomas by their molecular similarities allows more accurate prognostication and highlights possible treatment targets [3,4]. Another example of a molecular biomarker is *MGMT* promoter methylation. This gene encodes for the *MGMT* protein, which counteracts the effects of temozolomide, the main chemotherapy agent used in glioblastoma. When *MGMT* gene expression is silenced by promoter methylation, this confers an increased sensitivity to temozolomide and serves as a predictive biomarker for response [5].

Despite these advances in the molecular understanding of glioblastomas, the treatment protocols for glioblastomas have remained unchanged [2]. For this reason, considerable effort has been made to identify molecular targets. One such potential target is Prostate-Specific Membrane Antigen (PSMA). PSMA is a type II membrane protein composed of three components: an internal portion (19 amino acids), a transmembrane portion (24 amino acids), and a large extra-cellular portion (707 amino acids). It has known enzymatic activity, acting as a glutamate-preferring carboxypeptidase; however, its precise role remains unclear [6]. While PSMA was initially identified in prostate cancer cells, pan-tumor, and tumor-specific studies have identified its expression in a variety of other malignancies. For example, it has been shown to be expressed in the microvasculature of melanoma, lung, colon, and breast cancers [7,8]. In the clinical setting, PSMA expression in prostate cancer has been utilized for diagnostic purposes with gallium-68 (68 Ga) PSMA and therapeutically with Lutetium-177 (177 Lu) PSMA [9,10].

PSMA expression has also been demonstrated in glioblastoma. Using immunohistochemistry (IHC), Holzgreve et al. identified PSMA expression in 16 out of 16 glioblastoma patients. They also showed that PSMA IHC expression was associated with glioblastoma microvasculature and that this expression correlated negatively with survival [11]. It can be postulated that PSMA in the microvasculature plays a role in tumor neo-angiogenesis. Grant et al. suggested that angiogenesis associated with PSMA is independent of the Vascular Endothelial Growth Factor (VEGF) pathway [12]. The level of VEGF expression correlates with glioma grade and is thought to be the main promoter of angiogenesis in glioblastoma [13]. Anti-VEGF therapy, such as bevacizumab, is used in the management of glioblastomas. Although studies are yet to show bevacizumab has a survival benefit in glioblastoma [14], there is evidence of improvement in quality of life in its use, especially in the second line setting [15].

Given both PSMA and VEGF expression are possible targets in glioblastoma, the potential for their association needs to be explored. A previous study which included all glioma grades was unable to detect an association between PSMA and VEGF, but results may have been limited by a small sample size (*n* = 76) and a heterogenous population of low- and high-grade gliomas [16].

In this study, we evaluate the correlation between PSMA and VEGF expression in a cohort of *IDH1/2* wild-type glioblastomas and correlate it with known prognostic factors and outcomes.

## 2. Materials and Methods

### 2.1. Ethics

This was a retrospective observational study of adult patients with *IDH1/2* wild-type glioblastoma. The study was carried out in accordance with the approval and recommendations of the Northern Sydney Local Health District Human Ethics Committee.

### 2.2. Patient Cohort

The cohort consisted of 247 patients diagnosed with *IDH/1/2* wild-type glioblastoma between 2009 and 2014. Their archival formalin fixed paraffin embedded tumor samples were retrieved from the Kolling Institute tumor bank and Royal North Shore Hospital (RNSH) Department of Pathology. All included specimens complied with current WHO 2021 classification [4]. *IDH1* status was determined by routine IHC screening for *IDH1* p.R132H mutation performed by the RNSH Department of Pathology at the time of initial diagnosis (Clone H09 monoclonal antibody; Dianova, Hamburg, Germany). 

Clinical data, such as demographics, survival outcomes, and resection status, were retrieved. Relevant pathologic data, such as Ki-67 indices, was retrieved if performed on a clinical basis at the time of diagnosis. *MGMT* promotor methylation was not performed routinely during the years 2009 to 2014 of the cohort, but methylation status was recorded where available. 

### 2.3. Immunohistochemistry

Immunohistochemistry was performed on 4 µm sections of glioblastoma tissue microarrays, with 3 representative cores per patient. Antigen retrieval was performed in Tris/EDTA buffer, pH9 for 20 min at 95 °C. Sections were incubated with primary antibody for 1 h at room temperature, using either mouse anti-PSMA (Dako M3620, 1.12 µg /mL), mouse anti-CD31 (Dako M0823, 2.05 µg /mL), or rabbit monoclonal anti-VEGF (covering VEGF A, B and C variants) (Abcam ab52917, 1:100). CD31 immunostaining was included as a marker of endothelial cells. Negative isotype controls were included. After endogenous peroxidase blocking (0.3% H_2_O_2_, 5 min), sections were incubated for 15 min with Dako Mouse Linker (SM804) or Dako Rabbit Linker (K4003), followed by Dual Link-HRP (Dako K5007) for 30 min. ImmPACT NovaRED (Vector Lab, SK-4805) for 15 min was used for visualization. 

PSMA expression was examined for endothelial staining. PSMA endothelial cell staining was semi-quantitatively graded (0–3+) as follows: 0 = negative; 1+ = weak expression, <25% of endothelial cells and/or weak intensity; 2+ = moderate expression, 25–75% of endothelial cell staining and/or moderate to strong intensity; 3+ = high expression, >75% of endothelial cell staining and at strong intensity. VEGF vascular expression was scored by vascular endothelial intensity (0–2+) (See Figure 1).

Scoring was performed in a blinded fashion by 2 independent observers. PSMA was dichotomized into low (0–2+) and high (3+). VEGF was dichotomized into low (0–1+) and high (2+). To explore the effect of the PSMA threshold, a further analysis using a PSMA high cut-off of 2–3+ was also performed.

### 2.4. Statistical Analysis

Demographics, prognostic markers, and PSMA IHC expression were summarized with descriptive statistics. The distribution of patient features was assessed using either a two-tailed Student’s *t*-test or Chi^2^ analysis where appropriate. The levels of VEGF intensity were compared between PSMA high and low expressing glioblastomas using Chi^2^ analysis. The effect of PSMA expression on overall survival was summarized using Kaplan–Meier survival curves and compared using log-rank analysis. All the above analyses were conducted using GraphPad Prism version 9.

Multivariate analysis was conducted using multi-linear regression. The dependent variable was overall survival. The following independent variables were used: age, gender, extent of resection, Ki67 index, and PSMA vascular endothelial expression. Gender, the extent of resection, and PSMA expression were dichotomized into female/male, gross total/non-gross total resections, and PSMA expression 3+/non-3+, respectively. Validation of the multivariate analysis assumptions was assessed using analysis of residuals (variance of residuals and normalization assessed by visual inspection of relevant scatter plots). The correlation was assessed through correlation matrices to further explore how the PSMA high cut-off reflected VEGF expression. Multivariate analysis was conducted using IBM SPSS version 28.

## 3. Results

### 3.1. Cohort Demographics

Demographic and clinical features were statistically balanced for PSMA high vs. low expression, except for age (see Table 1). Of the 46 cases that were tested for *MGMT* methylation at the time of diagnosis, 14 were methylated (30.4%). PSMA high expression was observed in 35 patients (14.2%). In the PSMA low group, 54 patients (25.5%) had no expression, while both 1+ and 2+ scores had 79 patients (37.3%) each. 

### 3.2. Association of PSMA and VEGF Immunohistochemistry

The PSMA high cohort had a greater proportion of VEGF high expressing samples when compared with the PSMA low cohort (see Figure 2). In the PSMA high group, 47.1% of patients had high VEGF endothelial cell expression compared to 25.5% in the low VEGF endothelial cell expression (*p* = 0.001 by Chi^2^ test). VEGF status could not be performed in five patients (*n* = 1 in the high PSMA cohort and *n* = 4 in the low PSMA cohort).

When we reduced the PSMA high cut-off to 2+, the difference in endothelial VEGF expression remained statistically significant (*p* = 0.04 by Chi^2^ test). In this PSMA high cohort (2+ cut-off), 36.6% had high VEGF endothelial cell expressions compared to 25.5% of patients with low PSMA expression. 

### 3.3. Survival Outcomes

The median follow-up for survival analysis was 12 months (range 0.1–73 months), and no patients required censoring. The median overall survival (OS) was higher in the PSMA high cohort when compared to the PSMA low cohort (16.1 months vs. 10.8 months, *p* = 0.02; Figure 3). On multivariate analysis, there was no statistical difference in OS between PSMA high and low cohorts.

## 4. Discussion

This exploratory study demonstrates a potential positive correlation between PSMA and VEGF vascular endothelial expression. In total, 50% of glioblastomas with high PSMA expression (3+) also had high VEGF expression compared to 25% in the low PSMA group. These results remained significant when adjusting the PSMA cut-off to 2+. PSMA high expression had a potential positive correlation with OS univariate analysis, however, not on multivariate analysis. 

Some of these findings contradict the current literature. Our results contrast with that of Saffar et al., who did not demonstrate an association between PSMA and VEGF. Our study had a larger cohort (*n* = 247 compared to *n* = 72) and only included glioblastomas [16]. Holzgreve et al. found a negative association between PSMA IHC expression and OS; however, their sample size was small (*n* = 16) [11].

We acknowledge the limitations of our study. The retrospective nature of this review adds potential for bias, and future prospective studies are needed to validate our findings. Demographic details such as smoking status and alcohol intake were not available, limiting the description of the patient groups. However, through presenting the Eastern Cooperative Oncology Group (ECOG) status, it is hoped that an overall picture of the background health of each group can be appreciated. This cohort also predates the adoption of bevacizumab into routine clinical practice, and the widespread use of this anti-VEGF therapy may further alter the data in this population, which would not have been captured in our cohort. Given the relatively subjective measure of IHC staining, quantifying our findings with a more objective measure of expression, such as RNA sequencing, would be ideal. Unfortunately, there was not sufficient fresh frozen tissue available to achieve this with our historic samples. However, it is reassuring that the association between PSMA and VEGF IHC staining persisted when the high PSMA expression group was changed from 3+ only to 2–3+. There is also a degree of assumption that PSMA expression on IHC staining correlates with PSMA PET avidity. This being said, PSMA PET avidity is known to be strongly correlated with IHC expression in prostate cancer [17]. Furthermore, Holzgreve et al. demonstrated a case of PSMA-PET avidity in glioblastoma that was also positive on PSMA IHC staining [11]. 

Although the exploratory nature of our study prevents hard or clinically relevant conclusions, it does suggest avenues for future investigation. Initial studies should expand on the early data correlating glioblastoma PSMA IHC expression with PSMA PET avidity. This would, in turn, allow for biomarker studies. In addition, further larger studies using molecular assessments of PSMA-VEGF association, such as RNA sequencing, are required to provide confirmation of our findings. Depending on these additional translational studies, we may be able to postulate that the use of anti-VEGF therapy, such as bevacizumab, may be more effective in patients with increased PSMA expression. Therefore, if the significant sensitivity and specificity of PSMA-PET scans seen in prostate cancer [18] could be replicated in gliomas, PSMA PET scans could be a powerful predictive tool for bevacizumab response. Supporting this, there is early evidence regarding PSMA PET avidity in glioblastomas with reports of PSMA-PET scan avid gliomas found incidentally in patients with prostate cancer [19]. Furthermore, Verma et al. evaluated 10 patients with 68 Ga-PSMA PET/CT with suspected gliomas on recent imaging, with in vivo PSMA expression seen in all patients with glioma. They also showed that those patients with glioblastoma (*n* = 7) exhibited higher PSMA uptake than those with WHO grade 2 glioma (*n* = 3) [20].

If the above studies sufficiently characterize this association with PSMA and VEGF expression, this could eventually lead to therapeutic studies. Given that targeted therapies exist for VEGF and PSMA, future studies could explore the potential synergy of these treatments as a novel regimen for glioblastomas. For example, positive PSMA expression and correlation with PSMA-PET avidity in glioma may lead to significant opportunities in radiopharmaceuticals with or without traditional targeted therapies. The most commonly used example of this is 177 Lu–PSMA which has been successfully used in prostate cancer [9].

In conclusion, this study demonstrates a potential association between PSMA and VEGF vascular endothelial expressions in glioblastoma. PSMA expression had a potential positive correlation with OS. Further studies are required to investigate the role of PSMA expression in glioblastoma and its relationship to VEGF expression. 

## Figures and Tables

**Figure 1 biomedicines-11-01148-f001:**
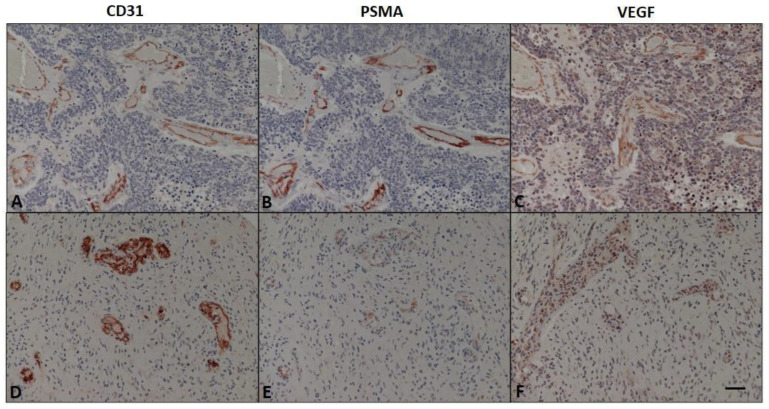
Immunohistochemical stains of vascular endothelial expression. (**A**,**D**): CD31 staining of vessels for reference; (**B**,**E**): High and low PSMA expression, respectively; (**C**,**F**): High and low VEGF expression, respectively. Scale bar = 50 μm.

**Figure 2 biomedicines-11-01148-f002:**
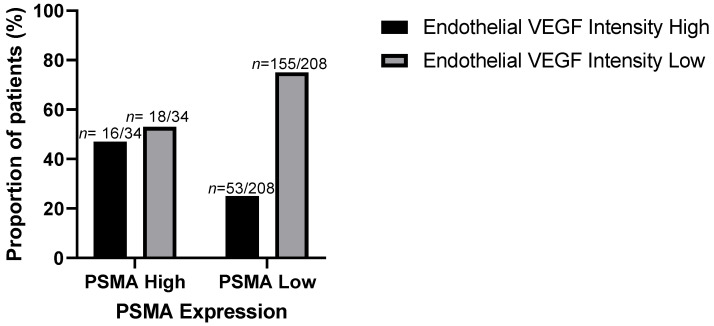
Proportion of VEGF vascular endothelial intensity in PSMA expression high and low groups.

**Figure 3 biomedicines-11-01148-f003:**
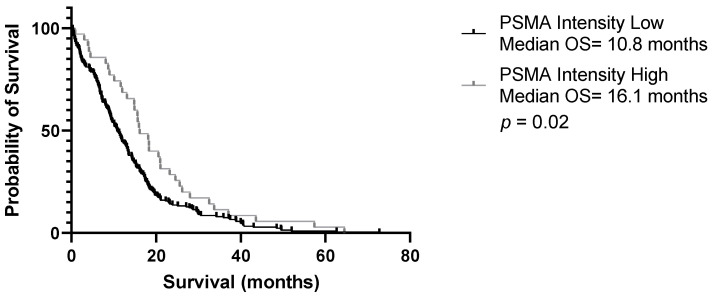
Survival curve for PSMA high expression vs. low expression.

**Table 1 biomedicines-11-01148-t001:** Patient characteristics. RT- radiotherapy; TMZ- temozolomide; ECOG- Eastern Cooperative Oncology Group.

	PSMA Expression High *n* = 35	PSMA Expression Low *n* = 212	*p* Value
Median Age (years, range)	61 (18–86)	66 years (17–87)	0.04
Gender (% female)	17 (48.6)	77 (36.3)	0.17
Resection status *n* (%) Gross total resection Subtotal resection Not reported	14 (40.0) 21 (60.0) 0	68 (32.1) 142 (67.0) 2 (0.9)	0.36
Median Ki67% (range) Not reported (*n*)	30 (10–90) 3	30 (5–90) 8	0.66
*MGMT* Promotor Methylation status *n* (%) Unmethylated Methylated Not performed/unknown	6 (17.1) 1 (2.9) 28 (80.0)	26 (12.3) 13 (6.1) 173 (81.6)	-
First line treatment *n* (%) Concurrent RT-TMZ followed by sequential TMZ Other regimen Not reported	23 (65.7) 12 (34.3) 0	105 (49.5) 98 (46.2) 9 (4.2)	0.13
ECOG Performance status *n* (%) 0 1 2 3 4 Not reported	14 (40.0) 12 (34.3) 5 (14.3) 2 (5.7) 0 2 (5.7)	50 (23.6) 73 (34.4) 48 (22.6) 18 (8.5) 1 (0.5) 22 (10.4)	-

## Data Availability

Data is not publicly available due to confidentiality issues.

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
