# Peer review of "PSMA Expression Correlates with Improved Overall Survival and VEGF Expression in Glioblastoma"

_biomedicines, 2023, doi:10.3390/biomedicines11041148_

Round 1
Reviewer 1 Report
The manuscript of Yuile et al reports studies on glioblastoma cancer. In particular, the authors highlight a positive correlation between PSMA and VEGF expression.
In my opinion, the manuscript is well written. I only ask to give more information on the population of individuals studied, considering the small number of this population (only 247).
In particular, a table with more information such as: average age, median, smokers or non-smokers, alcohol consumption, etc etc and if they have regressed pathologies. Anything that gives more information to statistically define the population under study.
Author Response
The manuscript of Yuile et al reports studies on glioblastoma cancer. In particular, the authors highlight a positive correlation between PSMA and VEGF expression.
In my opinion, the manuscript is well written. I only ask to give more information on the population of individuals studied, considering the small number of this population (only 247).
In particular, a table with more information such as: average age, median, smokers or non-smokers, alcohol consumption, etc etc and if they have regressed pathologies. Anything that gives more information to statistically define the population under study.
Response
Thank you for reviewing our manuscript. We agree more information to our Table 1 would add further context. Conventionally, smoking and alcohol status are not reported in glioblastoma studies as they do not impact tumorigenesis or molecular phenotypes.
Smoking status and alcohol intake are not routinely available nor recorded in the same manner across hospital records, making collation of this data difficult.
Recognising the reviewer’s desire to further define the clinical context we have collected the Eastern Cooperative Oncology Group (ECOG) Performance Status Score at the time of diagnosis. This is present for most patients and is recorded uniformly in patient notes.
This change has been added to Table 1.
We have also outlined the unavailable demographic features such as smoking status and alcohol intake in limitations- lines 221-224:
“Demographic details such as smoking status and alcohol intake were not available, limiting the description of the patient groups. However, through presenting ECOG status it is hoped that an overall picture of background health for each group can be appreciated.”
Reviewer 2 Report
The authors reported that PSMA expression in gilioblastoma correlates with prognosis for survival or with the level of VEGF expression.
The study was dichotomized into high and low PSMA expression, but the number of cases with high expression was relatively small. However, this may be an unavoidable part. Of course, as the authors write, forward-looking studies will be essential.
As for the data, there does not seem to be a major issue.
However, in general, the report is written too simply, and a more detailed description is needed. For example, PSMA is a topic of prostate cancer, but it seems to be reported to be expressed in glioma and other cancers as well, or is something being investigated clinically in each carcinoma? I would like to see a description.
Additionally, in Table 1, methylation of MGMT, etc. is noted, but it is just stated in the Methods column that it is not done in this cohort. However, if it is to be presented as part of the results, its clinical and molecular biological status should also be stated in the introduction, and the results should also be discussed.
It is not appropriate to note the first line treatment in Table 1 by reference number with the author's name.
The authors mention the term "correlations," but since these are basically IHC findings, PSMA and VEGF are only divided into two groups (high and low), respectively. It would be appropriate if at least a "correlation" could be found by gene expression or other quantitative numbers, but this is very rough. Any comments on this improvement?
Author Response
Thank you for your thorough review of our manuscript. We have endeavoured to address all of your comments as follows.
Comment 1
The study was dichotomized into high and low PSMA expression, but the number of cases with high expression was relatively small. However, this may be an unavoidable part. Of course, as the authors write, forward-looking studies will be essential.
Response 1
Thank you, we agree that forward-looking studies are required to expand on our findings.
Comment 2
As for the data, there does not seem to be a major issue.
However, in general, the report is written too simply, and a more detailed description is needed. For example, PSMA is a topic of prostate cancer, but it seems to be reported to be expressed in glioma and other cancers as well, or is something being investigated clinically in each carcinoma? I would like to see a description.
Response 2
We agree this is important to discuss. We have added further detail regarding the clinical interest of PSMA and how the initial focus of PSMA in prostate cancer has expanded to other tumour types. Lines 53-58:
“While PSMA was initially identified in prostate cancer cells, pan-tumour and tumour specific studies have identified its expression in a variety of other malignancies. For example, it has been shown to be expressed in the microvasculature of melanoma, lung, colon, and breast cancers (7,8).”
We have also expanded on our discussion as per your suggestions to increase the complexity of the study. In particular we have expanded the limitations section - lines 238-247:
“Although the exploratory nature of our study prevents hard or clinically relevant conclusions it does suggest avenues for future investigation. Initial studies should expand on the early data correlating glioblastoma PSMA IHC expression with PSMA PET avidity. This would in turn allow for biomarker studies. In addition, further larger studies using molecular assessments of PSMA-VEGF association, such as RNA sequencing, are required to provide confirmation of our findings. Depending on these additional translational studies we may be able postulate that the use of anti-VEGF therapy such as bevacizumab may be more effective in patients with increased PSMA expression.”
Comment 3
Additionally, in Table 1, methylation of MGMT, etc. is noted, but it is just stated in the Methods column that it is not done in this cohort. However, if it is to be presented as part of the results, its clinical and molecular biological status should also be stated in the introduction, and the results should also be discussed.
Response 3
In regards to MGMT status, in the methods we state it was not performed routinely by our institution’s pathologists, but was still performed in some instances. We appreciate this was confusing and to avoid misinterpretation we have amended the sentence- line 102-104:
“MGMT promotor methylation was not performed routinely during the years 2009 to 2014 of the cohort, but methylation status was recorded where available.”
As per your suggestions we have also amended our background to include a clinical and molecular context for measuring MGMT promoter methylation status. Lines 40-45:
“Another example of a molecular biomarker is MGMT promoter methylation. This gene encodes for the MGMT protein which counteracts the effects of temozolomide, the main chemotherapy agent used in glioblastoma. When MGMT gene expression is silenced by promoter methylation, this confers an increased sensitivity to temozolomide and serves as a predictive biomarker for response(5).”
Comment 4
It is not appropriate to note the first line treatment in Table 1 by reference number with the author's name.
Response 4
Thank you, we have changed the description of treatment in Table 1 from Stupp et al to the following:
“Concurrent RT-TMZ followed by sequential TMZ” with descriptions for RT and TMZ abbreviations in the Table 1 caption.
Comment 5
The authors mention the term "correlations," but since these are basically IHC findings, PSMA and VEGF are only divided into two groups (high and low), respectively. It would be appropriate if at least a "correlation" could be found by gene expression or other quantitative numbers, but this is very rough. Any comments on this improvement?
Response 5
We agree that a less subjective and more quantifiable method than IHC would be ideal when assessing PSMA and VEGF correlation. Unfortunately given the rarity of glioblastoma we were required to use archival FFPE and did not have sufficient matched fresh frozen samples to facilitate analysis through more precise means, such as RNA sequencing. With this in mind and given ultimately the target of interest is the PSMA protein, rather than mRNA expression, we felt IHC expression sufficient for this early study.
We have added this issue of quantifiability of IHC staining to the limitations sections. Lines 227-230:
“Given the relatively subjective measure of IHC staining, quantifying our findings with a more quantitative measure of expression such as RNA sequencing would be ideal. Unfortunately, there was not sufficient fresh frozen tissue available to achieve this with our historic samples.”
There is also some reassurance that we found a persisting statistically significant, but less pronounced difference when lowering our IHC threshold. We have attempted to discuss this further as per your recommendations. Lines 230-232:
“However, it is reassuring that association between PSMA and VEGF IHC staining persisted when the PSMA expression high group was changed from 3+ only to 2-3+.”
We have also highlighted the need for molecular analysis to verify our findings in the future directions section of our discussion- lines 242-247:
“In addition, further larger studies using molecular assessments of PSMA-VEGF association, such as RNA sequencing, are required to provide confirmation of our findings. Depending on these additional translational studies we may be able to postulate that the use of anti-VEGF therapy such as bevacizumab may be more effective in patients with increased PSMA expression.”
We have changed “positive correlation” to “potential positive correlation” when referring to correlation between VEGF and PSMA IHC.
For example- Lines 206-207:
“This exploratory study demonstrates a potential positive correlation between PSMA and VEGF vascular endothelial expression.”
Reviewer 3 Report
This is a very good study investigating the association between PSMA and VEGF expression in the neo-vasculature of glioblastoma.
The Authors performed a very sound analysis and the paper is well written. I am afraid, however, that several assumptions are overstated, given the exploratory nature of this study (recognized by Authors themselves).
e.g., in the abstract "we demonstrate an association between PSMA and VEGF expression in glioblastoma neo-vasculature". And also the Discussion presents several cases of overstatements.
can the Authors tone down the text as necessary?
Author Response
Comment
This is a very good study investigating the association between PSMA and VEGF expression in the neo-vasculature of glioblastoma.
The Authors performed a very sound analysis and the paper is well written. I am afraid, however, that several assumptions are overstated, given the exploratory nature of this study (recognized by Authors themselves).
e.g., in the abstract "we demonstrate an association between PSMA and VEGF expression in glioblastoma neo-vasculature". And also the Discussion presents several cases of overstatements.
can the Authors tone down the text as necessary?
Response
Thank you for your review of our manuscript. We have attempted to modify our assumptions to be more in line with our exploratory study. As per your recommendations, we have made the following changes.
We have amended the sentence in the abstract “we demonstrate an association between PSMA and VEGF expression in glioblastoma neo-vasculature” to the following- lines 21-22:
“Our study suggests the potential for an association between PSMA and VEGF expression in glioblastoma neo-vasculature”.
We have changed “positive correlation” to “potential positive correlation” when referring to a correlation between VEGF and PSMA IHC.
For example- lines 29-30
“Conclusion: We found a potential positive correlation between PSMA and VEGF expression.”
Lines 206-207:
“This exploratory study demonstrates a potential positive correlation between PSMA and VEGF vascular endothelial expression.”
Lines 275-276:
“In conclusion, this study demonstrates a potential association between PSMA and VEGF vascular endothelial expressions in glioblastoma.”
We made the following changes to future directions- lines 238-247:
“Although the exploratory nature of our study prevents hard or clinically relevant conclusions, it does suggest avenues for future investigation. Initial studies should expand on the early data correlating glioblastoma PSMA IHC expression with PSMA PET avidity. This would in turn allow for biomarker studies. In addition, further larger studies using molecular assessments of PSMA-VEGF association, such as RNA sequencing, are required to provide confirmation of our findings. Depending on these additional translational studies we may be able postulate that the use of anti-VEGF therapy such as bevacizumab may be more effective in patients with increased PSMA expression.”
We have removed the following paragraph as we felt it placed too much emphasis on our findings, without acknowledging their exploratory nature- lines 268-272:
“Given our finding of a positive correlation, between PSMA and VEGF expression, it is not known how anti-VEGF therapy, which is in commonly used in recurrent glioblastoma currently, will affect PSMA PET avidity in PSMA expressing glioblastoma. Future functional PET imaging studies can investigate this relationship. This would then help determine sequencing of anti-VEGF and 177Lu-PSMA therapy.”
Round 2
Reviewer 2 Report
Now, authors modified their manuscript according to the reviewers' comments adequately.